# Polymorphic Appetite Effects on Waist Circumference Depend on rs3749474 CLOCK Gene Variant

**DOI:** 10.3390/nu12061846

**Published:** 2020-06-21

**Authors:** Isabel Espinosa-Salinas, Rodrigo San-Cristobal, Gonzalo Colmenarejo, Viviana Loria-Kohen, Susana Molina, Guillermo Reglero, Ana Ramirez de Molina, J. Alfredo Martinez

**Affiliations:** 1IMDEA-Food Institute, CEI UAM+CSIC, 28049 Madrid, Spain; rodrigo.sancristobal@imdea.org (R.S.-C.); gonzalo.colmenarejo@imdea.org (G.C.); viviana.loria@imdea.org (V.L.-K.); susana.molina@imdea.org (S.M.); guillermo.reglero@imdea.org (G.R.); jalfredo.martinez@imdea.org (J.A.M.); 2Institute of Food Science Research (CIAL) CEI UAM+CSIC, 28049 Madrid, Spain; 3Center for Nutrition Research (CIN), Navarra Institute for Health Research (IdiSNA), 31008 Pamplona, Spain; 4Center of Biomedical Research in Physiopathology of Obesity and Nutrition (CIBEROBN), Institute of Health Carlos III, 28029 Madrid, Spain

**Keywords:** chronobiology, biorhythm genes, obesity, metabolic syndrome, polymorphism, interaction

## Abstract

Chronobiological aspects controlled by *CLOCK* genes may influence obesity incidence. Although there are studies that show an association between the expression of these genes and energy intake, waist circumference or abdominal obesity phenotypes, interactions with appetite have been insufficiently investigated in relation to chrononutrition. The objective was to identify interactions between *CLOCK* genetic variants involved in appetite status. A total of 442 subjects (329 women, 113 men; aged 18 to 65 years) were recruited. Anthropometric, dietary and lifestyle data were collected by trained nutritionists. Participants were classified according to their appetite feelings with a Likert scale. Multiple linear regression models were used to examine associations of the type genotype x appetite status on adiposity-related variables. *p* values were corrected by the Bonferroni method. A significant influence was found concerning the effects of appetite on waist circumference with respect to rs3749474 *CLOCK* polymorphism (*p* < 0.001). An additive model analysis (adjusted by age, gender, exercise and energy intake) showed that risk allele carriers, increased the waist circumference around 14 cm (*β* = 14.1, CI = 6.3–22.0) by each increment in the level of appetite. The effects of appetite on waist circumference may be partly modulated by the rs3749474 *CLOCK* polymorphism.

## 1. Introduction

*CLOCK* genes exert an influence on weight regulation mediated by brain areas, that are likely to be involved in the control of relevant endocrine functions through anorexic and orexigenic hormones and neurotransmitters [1]. This complex controlling system depends on the regulation of *CLOCK* genes by transcriptional mechanisms [2].

Current available findings have proven an active circadian clock located in adipocytes, where genes expressed in the adipose tissue are under the influence of cyclic systems [3,4]. Investigations in recent years have shown that the *CLOCK* gene (Circadian Locomotor Output Cycles Kaput) is implicated in metabolic dysfunctions. However, more research about the biological rhythms control is needed [5]. Several studies suggest adipocyte differentiation and proliferation as well as endocrine factors synthesis in the adipose tissue, are regulated by circadian rhythms, implicating hormonal and neuronal signals [6]. Thus, altered functions of the *CLOCK* gene could have negative effects on fat accumulation or mobilization in adipocytes as well as on healthy eating behaviors [7]. 

Body fat location in humans is considered as important as the total fat mass content concerning metabolic complications [8]. Thus, it has been widely demonstrated that visceral adipose tissue is a relevant predictor of adverse metabolic events, whereas subcutaneous adipose tissue appears not to be so harmful on health status [9]. In fact, the increased intra-abdominal fat depot is recognized as a risk factor for a dysfunctional state in insulin-sensitive tissues leading to the risk of later developing type 2 diabetes, cardiovascular disease and metabolic syndrome [9]. Previous studies have reported associations of *CLOCK* genetic variants (rs4864548, rs3736544, rs1801260 or rs3749474, rs4580704, rs1801260 haplotypes) with waist circumference [10,11]. More specifically, it was found that minor allele T carriers of rs3749474 *CLOCK* had significantly higher waist circumference, weight and BMI (Body Mass Index) than CC subjects [5]. The allele and genotype frequencies of the reference polymorphism (rs3749474) are C: 66%, T: 34% and C/C: 43% C/T: 45% T/T: 12% in the European population (Ensembl GRCh38). In turn, polymorphisms located within the 3′–UTR of the mRNA, can affect the functionality of the mRNA, as seen in the case of rs3749474 and rs1801260 [5,12,13]. 

On the other hand, some obesity treatment approaches are being focused on the regulation of appetite and energy consumption [14]. From a physiological point of view, one of the main signals of appetite is the timing of food intake during the day [1,4,15]. To date, the research has evidenced an association concerning minor allele carriers of rs3749474 *CLOCK* and energy intake [12]. However, there are not appropriate studies that report direct associations between appetite and *CLOCK* polymorphisms.

Analyzing the effects of *CLOCK* variants on adiposity markers can contribute to increasing the knowledge about the regulatory mechanisms that depend on biorhythm regulatory genes [7]. Therefore, the aim of this study was to identify novel interactions between *CLOCK* genes involved in the development of obesity and associated comorbidities, according to appetite degree, since although there are studies that show an association between these processes, the scientific evidence about the interactions between both factors is scarce.

## 2. Materials and Methods

### 2.1. Subjects and Study Protocol

Subjects recruitment for this observational study was achieved through the Platform for Clinical Trials in Nutrition and Health (GENYAL), belonging to IMDEA Food Institute (Madrid, Spain). Inclusion criteria for recruitment were: free-living adults aged 18 to 70 years that had to give a written informed consent to be contacted to perform clinical trials and nutritional intervention studies. Exclusion criteria were: to suffer from any serious illness (kidney or liver diseases, or other condition that affects lifestyle or diet), to present dementia or impaired cognitive function, and to be pregnant or breastfeeding. A total of 1249 subjects were contacted, 954 of whom were recruited. Of these, a total of 557 participants were finally selected. Due to uncompleted questionnaires, 487 samples were left for analysis. The sample size of the variables ranges from 437 to 487 due to some missing data. No imputation was performed. The sample size was shown considering all subjects with age, sex and genotyping data for at least some one of the three polymorphisms (*n* = 442). CONSORT flow diagram is shown as Appendix A. The study was conducted according to the guidelines laid down in the Declaration of Helsinki and all procedures involving human subjects were approved by the Research Ethic Committee of Autonomous University of Madrid (CEI 27–666). Written informed consent was obtained from all subjects before starting the intervention. 

### 2.2. Nutritional Assessments

Anthropometric and body composition variables were evaluated by standard validated techniques [16,17]. Height was measured using a Leicester stadiometer (Biological Medical Technology SL, Barcelona, Spain). Body weight, body mass index (BMI), fat mass, lean mass, visceral fat classification and basal metabolism were assessed using the body composition monitor BF511 (Omron Healthcare UK, LT, Kyoto, Japan). Waist and hip circumferences were determined using a Seca 201 non-elastic tape (Quirumed, Valencia, Spain). All anthropometric parameters were measured by trained nutritionists on an empty bladder with a minimum of two hours fasting, as described elsewhere [18]. 

To measure appetite degree, a Likert scale with 5 points was used (where 1–2 means a low rate of appetite, 3 an intermediate level and 4–5 a lot of appetite). This question was asked by trained nutritionist always at the research center and when the subjects had been fasting for at least two hours following published protocol [19,20]. The participants responded according to their habitual appetite status with a commonly used question by Spanish population to refer to how you feel about your eating and feeding. Then, since it is a semi-quantitative method, the data were merged in 3 categories in order to optimize the statistical analysis. This scale was categorized in low appetite, medium appetite and high appetite with 1–2, 3, and 4–5 responses, respectively. In order to evaluate exercise levels, it was assessed the frequency of exercise considering it as an increase in body mobility being a planned and repetitive activity [21]. The Minnesota Leisure Time Physical Activity Questionnaire has been applied to assess the amount of physical activity in (kcal/day), considering physical activity as any voluntary movement produced by the muscles and resulting in the expenditure of energy [21].

A 72–h food record validated for the Spanish population [22], was collected from all participants. Participants received training to record the questionnaire. The DIAL (2.16 version, Alce Ingeniería) Software [23] was used to analyze the energy, macro and micronutrients intake from the dietary records [24,25].

In order to evaluate physical activity levels, participants had to indicate how many days per week they did physical activity as described previously [26]. Those subjects that reported at least 1 day/week of physical activity were considered active, while the remaining were considered sedentary. Moreover, the Minnesota Leisure Time Physical Activity Questionnaire (MLTPAQ) was administered to quantitatively measure the average physical activity practice (kcal/week) by the subjects according to published criteria [27,28]. Thereafter, using a Compendium of physical activities [29] and the American Heart Association Guidelines [30], volunteers were also classified into sedentary and physically active groups. Energy Expenditure in Physical Activity was estimated as follows: I × N × T; where “I” represents the level of intensity for each physical activity in kilocalories/min; “N”, the number of times that physical activity was reported; and “T”, the time in minutes spent in each session as described elsewhere [26].

### 2.3. DNA Extraction and Genotyping 

*CLOCK* rs3749474, rs1801260 and rs4580704 genetic variants were screened following previously published procedures [5,31]. Blood samples were taken and kept at −80 °C until ADN extraction. Genomic DNA from each participant was isolated from 300 μL of total blood using the QIAamp DNA Blood Mini Kit (Qiagen Sciences, Inc, Germantown, MD, USA) and recovered in 100 mL of nuclease-free water. Its concentration and quality were then measured in a nanodrop ND-2000 spectrophotometer (ThermoScientific, Waltham, MA, USA). The mean concentration of the samples was 80 to 90 ng/mL. Genotyping was performed using the QuantStudio_ 12 K Flex Real-Time PCR System (Life Technologies Inc., Carlsbad, CA, USA) with a TaqMan OpenArray plates following manufacturer’s instructions (Real-Time PCR Handbook and education center of Applied Biosystem) [32]. The results were analyzed using TaqMan Genotyper software [33]. The proportion of genotypes not passing the quality threshold was < 5%. The samples were made in duplicate, resulting more 99% of the genotyping results were technically valid.

### 2.4. Statistical Analyses

Statistical analyses were performed with R software, version 3.4.1 (R Foundation for Statistical Computing, Vienna, Austria) [34]. Deviations from Hardy-Weinberg equilibrium of genotype frequencies at individual *loci* were assessed using standard *χ*^2^ tests. Descriptive analyses were implemented for different continuous and categorical variables by sex. The *p*-values were obtained using Student’s t-test for continuous variables, and Fisher exact test for categorical variables. In turn, associations between 3 *CLOCK* (rs3749474, rs1801260 and rs4580704) polymorphism and studied variables, were modeled through linear models. Interaction models of the type *CLOCK* polymorphism*appetite grade in the prediction of obesity variables were created by using linear models. Three genetic models were considered, namely: additive, dominant and codominant, and 3 *CLOCK* polymorphisms related to biorhythms were analyzed: rs3749474, rs1801260 and rs4580704. Different adjustment variables were considered. The *p* values were corrected for multiple comparisons (Bonferroni) in all the cases following stepwise criteria. Significance level was set to α = 0.05, and bilateral tests were considered. 

## 3. Results

A total of 442 subjects (329 women and 113 men, with mean and ± SD age of 37.63 ± 12.38 years) were analyzed, considering the sample size, which included all subjects with age, sex and genotyping data for at least one of the three polymorphisms. Anthropometric measures, dietary intake, dietary habits, physical activity characteristics and frequency of studied genotypes, according to sex were completed (Table 1). Statistically significant differences concerning sex for dietary intake and appetite degree and physical activity were found (Table 1). Genotype distribution for *CLOCK* rs3749474, rs1801260 and rs4580704 was in accordance with the Hardy–Weinberg equilibrium (*p* = 0.483; *p* = 0.455 and *p* = 0.404, respectively).

Genotype association analyses were performed for *CLOCK* rs3749474, rs1801260 and rs4580704 considering anthropometric, dietary intake, dietary habits and physical activity characteristics, where no significant differences were found for either genotype adjusted by the Bonferroni method (Table 2).

*CLOCK*s interactions with appetite grade in the prediction of selected variables (BMI, fat mass, visceral fat and waist circumference) were subsequently analyzed. Among the three genetic models evaluated (additive, dominant and codominant), the best fit was obtained with the additive model (in which each copy of T modifies the risk by an additive amount and therefore T/T homozygotes have twice the risk of C/T heterozygotes). Of all the studied variables, statistically significant interactions were found for *CLOCK* rs3749474 risk allele in the prediction of waist circumference. The effect of the appetite level (low < medium < high) on the waist circumference diameter was studied according to the genotypes using marginal models. In addition, the analysis was performed with three different adjustments: age and gender (model 1), energy intake, age and gender (model 2), and exercise, energy intake, age and gender (model 3) as reporting in Table 3.

Statistically significant differences of waist circumference according to *CLOCK* rs3749474 and appetite classification were found for the three models (*p* ≤ 0.001) adjusted by age and gender (model 1), energy intake, age and gender (model 2) and exercise, energy intake, age and gender (model 3). In the model 1 (Table 3), in minor genotype carriers (T/T), the waist circumference increased by around 16 cm (*β* = 15.7, CI = 7.9–23.5) with each unit of increment in the degree of appetite (low < medium < high). However, in the common genotype carriers (C/C) the waist circumference only increased by around 3 cm (*β* = 3.1, CI = 0.3–5.8). In the model 2 and 3, similar results were obtained, since in T/T carriers, the waist circumference increased by about 14 cm (model 2: *β* = 14.0, CI = 6.0–22.0; and model 3: *β* = 14.1, CI = 6.3–22.0), in comparison with C/C carriers, who had an increase in waist circumference of only around 3 cm. 

Mean measures of waist circumference according to *CLOCK* rs3749474 genotype and appetite degree adjusted by exercise, energy intake, age and gender (model 3) are shown in Figure 1.

Subsequently, the data were reanalyzed in another study with a greater number of polymorphisms to screen other possible interactions. The results again were statistically significant (Model 1: *p* interaction = 0.029; Model 2: *p* interaction = 0.016; Model 3: *p* interaction = 0.008). In this case, Bonferroni correction criteria were used given the number of polymorphisms and dietary variables studied (adjusted *p* values were based on 81 tests involving the nine studied polymorphisms and nine dietary variables).

## 4. Discussion

The present study was designed to identify interactions between *CLOCK* genetic variants with appetite status. With regards to anthropometric measurements, a relevant finding that emerged from this study was a directly proportional association between the minor T allele of the *CLOCK* rs3749474 variant and the waist circumference, for each increase in the grade of appetite (additive model). In this respect, homozygous variant (T/T) carriers showed the greatest increase in waist diameter (about 14 cm) by each increment in the level of appetite (low < medium < high). 

Although evidence of associations between the *CLOCK* genes and waist circumference have been demonstrated earlier, no interactions with the grade of appetite have been reported before. On this basis, the present study may constitute an advance to understand the influence of the *CLOCK* genes on weight control. In this sense, this study suggests that the genetic component could determine the effect that the level of appetite has on cardiometabolic markers such as waist circumference. The fact of finding differences between the genotypes has the potential to help to focus nutrition strategies on target population with certain genetic predisposition in the future. However, more research on this topic needs to be undertaken before the interaction between appetites, *CLOCK* polymorphism and waist circumference is more clearly understood. At the same time, our study used the waist diameter as an indicator of visceral fat. Although waist circumference cannot discriminate between visceral and subcutaneous fat tissue, visceral fat tends to be directly proportional to waist diameter [35,36]. However, future studies could be developed using more direct techniques to measure visceral fat (e.g., by Dual-energy X-ray Absorptiometry). One additional limitation of this study may be the measurement of appetite level. Measuring appetite level is a rather subjective parameter. Therefore, more studies are required to confirm the results obtained in this study.

Other researchers have also found associations, based on visual analogue scales, with respect to other polymorphisms (such as rs9939609 FTO and rs12970134 near MC4R) and the level of appetite [31,37,38]. In the case of rs9939609 FTO they evaluated the influence of this single-nucleotide polymorphism on appetite, ghrelin, leptin, interleukin 6 (IL6), tumor necrosis factor α (TNFα) levels and food intake [37]. For rs12970134 near MC4R they associated the influence of this polymorphism on appetite and beverage intake [31]. Therefore, future studies with *CLOCK* polymorphisms could also include the analysis of interactions with genes that encode neurotransmitters, cytokines, palatable food, or hedonic eating parameters. The current analyses could be complemented by including appetite-related hormones such as ghrelin or leptin measurements would render this research much valuable, however, this could not be carried out due to insufficient blood sample.

From the descriptive analysis, a significant difference between men and women in the appetite rate was appreciated. Simple sugars were included in the total carbohydrates. However, we considered separating out simple sugars since they may show a different outcome than overall carbohydrates given their role on sweetness and satisfaction as well as potentially on appetite [39]. This is comparable with earlier observations, which showed that appetite ratings differed according to age, gender, and physical activity [40,41]. However, the interaction analysis in our study was adjusted for age, sex and energy intake, so these variables did not apparently influence the results. 

In turn, the expression of these genes may influence the regulation of lipid and carbohydrate homeostasis as well as the adipose tissue and abdominal fat content [42]. On this basis, the *CLOCK* genes have been shown to influence the regulation of the expression of adipocytokines such as adiponectin, resistin and leptin, the synthesis of which daily vary throughout the 24–h cycle [43].

Thus, these genes have been associated with diseases such as obesity and metabolic syndrome. Specifically, it has been reported that the rs3749474 *CLOCK* variant generates a change in the structure of the messenger RNA in such a way that the level of expression is reduced [12,44]. This shift has been associated with a higher total energy intake as well as a higher fat intake, resulting in and increased abdominal obesity in subjects carrying the variant [12,44]. In particular, the existing literature reveals that carriers of the T allele present a greater risk of weight gain [12,44]. 

When analyzing the results of this interaction in more detail, it can be noted that the T/T carriers + low appetite degree had a lower average waist diameter than the other genotypes carriers (T/T: 69.00 ± 0.00 cm; C/T: 75.33 ± 12.64 cm; C/C: 85.86 ± 15.77 cm). Conversely, subjects with a high appetite degree + T/T carriers showed much higher levels of appetite compared to carriers of the other genotypes (T/T: 95.35 ± 18.27 cm; C/T: 91.06 ± 14.49 cm; C/C: 89.42 ±13.45 cm). These results suggest that it may be especially interesting to investigate how to reduce the degree of appetite in T/T carriers, as the impact may be particularly beneficial for a population with such genotype.

Interestingly, our results also found an interaction concerning to waist circumference: carriers of the T/T genotype increased by about 14 cm for each increase in the appetite rate (C/T: 7 cm and C/C: 3 cm). In addition to weight gain, previous studies have reported the association of the biorhythm genes with abdominal fat content [43,45]. Therefore, alterations in the *CLOCK* gene could be linked with a greater accumulation of visceral fat. Abdominal fat generates a multitude of adverse metabolic processes associated with inflammation and chronic disease [35,36]. This outcome explains under an additive model that subjects with T/T genotype may be at increased risk for the etiopathogenesis of diseases related to metabolic syndrome and abdominal obesity (and this risk seems to grow with each increase in the level of appetite according to the results found). In turn, scientific literature has shown that in the analysis by haplotypes rs3749474, rs45807041 and rs1801260, carriers of the haplotype CGA had a lower BMI, weight, waist circumference than did noncarriers [11]. This evidence corroborates our results and in turn indicates the possibility that the other polymorphisms (rs45807041 and rs1801260) affect body weight and waist diameter together.

The adjustment for multiple comparisons took into account 3 SNPs * 1 Appetite interaction test. The unadjusted *p* values were: *p* = 0.00036 (Model 1), *p* = 0.00019 (Model 2) and *p* = 0.000095 (Model 3). In case of making the correction taking into account the rest of the dependent variables, *p* adjusted interaction values continued to be equally significant (Model 1: *p* = 0.004; Model 2: *p* = 0.002 Model 3: *p* = 0.001). Although we had a greater number of adjustments in the confirmatory analysis (81 tests), in this case the dependent variables were not taken into account since we considered not necessary to perform further correction analyses involving multiple comparisons. Failure to address multiple comparisons appropriately can introduce excess false positive results and make subsequent studies following up those results inefficient [46].

In addition to an increased energy intake, associations to *CLOCK* genes variants have shown preferences for eating less healthy, higher in calories and higher in glycemic index foods [47]. Thus, sleep-deprived subjects showed a greater stimulation response to intake, when they were exposed to unhealthy foods [47]. By contrast, an increase in abdominal fat and metabolic alterations were not evident in rs3749474 variant carriers who had a healthier diet (higher than average olive oil consumption) [42]. On this basis, it can be assumed that hormonal changes caused by *CLOCK* gene influence eating behavior, energy metabolism and fat tissue. Therefore, eating satiating foods could be a good strategy to control the degree of appetite and prevent the accumulation of visceral fat in this case.

## 5. Conclusions

The impact of appetite on waist circumference is partially modulated by the variability in the rs3749474 *CLOCK* polymorphism. Appetite control strategies in T/T carriers of this variant could be specifically useful to prevent metabolic alterations characteristic of the increase in abdominal fat. Such approach could include adding healthier foods to the diet. Further studies on the interaction between *CLOCK* genes and appetite levels would be interesting to confirm this novel association for personalized precision nutrition implementation.

## Figures and Tables

**Figure 1 nutrients-12-01846-f001:**
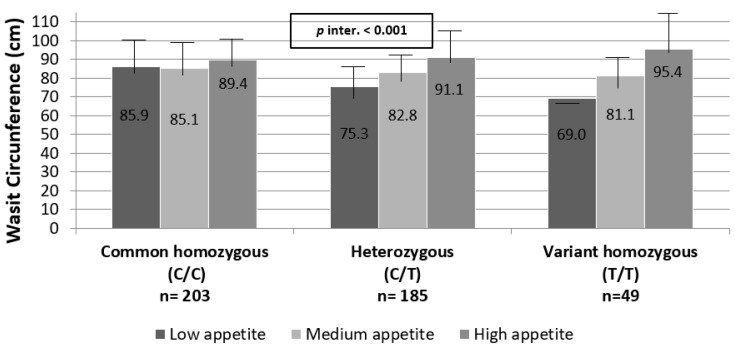
Waist circumference according to *CLOCK* rs3749474 and appetite classification. *p* interaction in additive model adjusted by exercise, energy intake, age and gender (Model 3).

**Table 1 nutrients-12-01846-t001:** General characteristics of the studied sample (X ± SD; %).

Variables	Total (*n* = 442)	Female (*n* = 329)	Male (*n* = 113)	*p*
Body weight (kg)	72.1 (±15.6)	67.7 (±13.5)	84.3 (±14.7)	<0.001
Height (cm)	166 (±9)	162 (±6)	176 (±6)	<0.001
BMI (kg/m^2^)	26.1 (±4.9)	25.7 (±4.8)	27.2 (±4.8)	0.002
Fat mass (%)	33.8 (±10.0)	37.2 (±8.4)	24.5 (±8.0)	<0.001
Lean mass (%)	28.9 (±5.8)	26.5 (±3.5)	36.0 (±5.0)	<0.001
Visceral fat category Normal	79	88	54	<0.001
High	14	10	25	
Very High	7	2	21	
Waist circumference (cm)	87.0 (±14.5)	84.4 (±13.7)	94.4 (±4.1)	<0.001
Waist category (%) No risk	50	50	52	0.679
Risk	50	50	48	
Waist-Hip ratio	0.8 (±0.1)	0.8(±0.1)	0.89 (±0.1)	<0.001
Estimated basal metabolism (Kcal/day)	1485 (±244)	1384 (±152)	1804 (±203)	<0.001
Energy (TCV: kcal/day)	2115 (±712)	2037 (±719)	2325 (±650)	<0.001
CHO (TCV %)	38.3 (±6.5)	38.5 (±6.3)	37.8 (±7.0)	0.249
Simple sugars (VTC %)	17.6 (±5.2)	17.9 (±5.2)	16.5 (±5.1)	0.004
Proteins (TCV %)	17.3 (±3.4)	17.3 (±3.5)	17.2 (±3.3)	0.999
Fats (TCV %)	40.0 (±6.4)	40.0 (±6.4)	40.0 (±6.4)	0.922
Appetite category (%) Low	6	7	2	0.006
Medium	42	45	35	
High	52	48	63	
Meals/day frequency category (%) ≥4	64	67	54	0.008
<4	36	33	46	
Exercise (%) Inactive	31	33	24	0.075
Active	69	67	76	
Energy expenditure physical activity/week (kcal/week)	2207 (±1917)	2038 (±1676)	2701 (±2434)	0.015
rs3749474 (%) C/C	47	44	52	0.299
C/T	42	44	40	
T/T	11	12	8	
rs1801260 (%) A/A	54	55	52	0.764
A/G	40	39	41	
G/G	6	6	7	
rs4580704 (%) C/C	38	38	36	0.743
C/G	46	46	46	
G/G	16	16	18	

Continuous variables, X ± SD; categorical variables, %. The sample size of the variables ranges from 437 to 487 due to some missing data. No imputation was performed. The sample size was shown considering all subjects with age, sex and genotyping data for at least one of the three polymorphisms. BMI, Body mass index; CHO, carbohydrates; TCV, total caloric value. Inactive, 0 times of physical activity performance per week. Active, one or more times of physical activity performance per week. Significance level *p* ≤ 0.05.

**Table 2 nutrients-12-01846-t002:** Association of *CLOCK* SNPs with studied variables (X ± SD; %).

Variables	rs3749474					rs1801260								rs4580704		
	C/C		C/T	T/T	*p*	A/A	A/G		G/G		*p*		C/C	C/T	T/T	*p*
	*n* = 203		*n* = 185	*n* = 49		*n* = 243	*n* = 178		*n* = 27				*n* = 173	*n* = 210	*n* = 75	
Body weight (kg)	71.8 (±15.0)		72.5 (±15.3)	74.6 (±18.5)	0.099	72.4 (±15.4)	72.3(±15.6)		70.6 (±16.9)		1		72.4 (±16.8)	72.0 (±15.5)	71.3(±12.4)	0.866
Height (cm)	166 (±9)		167 (±9)	166 (±10)	0.032	166.3 (±8.9)	165.6 (±8.9)		165.8 (±9.8)		0.392		166.0 (±9.3)	165.9 (±8.9)	165.6 (±8.0)	1
BMI (kg/m^2^)	26.2 (±4.9)		26.1 (±4.7)	26.98 (±5.8)	0.834	26.2 (±5.0)	26.3 (±4.8)		25.5 (±4.5)		1		26.1 (±4.9)	26.1 (±5.1)	26.1 (±4.4)	1
Fat mass (%)	33.2 (±9.8)		34.3 (±9.7)	35.8 (±11.5)	1	33.7 (±10.6)	34.3 (±9.3)		32.5 (±7.8)		1		34.5 (±9.5)	33.5 (±10.1)	33.2 (±10.7)	1
Lean mass (%)	29.2 (±5.9)		28.6 (±5.6)	28.8 (±6.3)	1	29.2 (±6.1)	28.5 (±5.5)		29.1 (±4.9)		0.759		29 (±5.5)	29.0 (±5.9)	29.6 (±6.4)	0.908
VFC Normal	75		83	81	1	81	76		82		1		82	78	78	1
High	15		14	15		14	16		11				14	14	15	
Very High	10		3	4		5	8		7				4	8	7	
WC (cm)	87.2 (±14.0)		86.8 (±13.9)	88.9 (±17.7)	0.53	86.7 (±14.4)	87.7 (±14.3)		86.0 (±14.0)		1		87.1 (±14.9)	87.0 (±14.6)	86.2 (±12.3)	1
Waist class No risk	49		51	50	1	53	48		54		1		51	50	51	1
Risk	51		49	50		47	52		46				49	50	49	
Waist-Hip ratio	0.83 (±0.1)		0.82 (±0.1)	0.83(±0.1)	1	0.82 (±0.1)	0.83 (±0.1)		0.82 (±0.1)		1		0.82 (±0.1)	0.83 (±0.1)	0.82 (±0.1)	1
EBM (Kcal/day)	1492 (±244)		1487 (±241)	1501 (±250)	0.183	1489 (±239)	1488 (±244)		1492 (±290)		1		1485 (±254)	1487 (±241)	1479 (±217)	0.945
Energy (TCV: kcal/day)	2049 (±548)		2214 (±909)	2113 (±592)	0.342	2069 (±566)	2226 (±956)		2128 (±521)		0.433		2271 (±974)	2037 (±502)	2079 (±639)	0.049
CHO (TCV %)	37.6 (±6.4)		38.5 (±6.3)	38.8 (±6.7)	0.622	38.1 (±6.2)	38.4 (±6.8)		39.9 (±7.1)		0.913		39.1 (±6.9)	38.1 (±6.0)	37.0 (±6.2)	0.145
Simple sugars (VTC %)	16.9 (±4.5)		17.9 (±5.6)	18.2 (±5.2)	0.211	17.5 (±5.0)	17.6 (±5.4)		18.3 (±5.1)		1		18.2 (±5.8)	17.4 (±4.7)	16.7 (±4.9)	0.134
Proteins (TCV %)	17.3 (±3.3)		17.2 (±3.5)	16.8 (±2.8)	1	17.1 (±2.9)	17.6 (±3.7)		16.6 (±4.0)		1		17.0 (±3.6)	17.4 (±3.2)	17.2 (±2.9)	1
Fats (TCV %)	40.7 (±6.3)		39.8 (±6.0)	40.0 (±5.6)	0.888	40.3 (±5.7)	39.9 (±6.6)		38.5 (±7.4)		0.778		39.4 (±6.5)	39.9 (±5.9)	41.8 (±5.7)	0.096
Appetite class Low	6		8	4	0.803	6	7		4		0.534		7	5	9	1
Medium	45		37	38		38	44		54				43	41	41	
High	49		55	58		56	49		42				50	54	50	
Meals frequency class (Meals/day) ≥4	61		64	69	1	64	62		65		1		66.	62	65	1
<4	39		36	31		36	38		35				34	38	35	
Exercise Inactive	29		28	31	1	30	30		19		1		28	30	29	1
Active	71		72	69		70	70		81				72	70	71	
EEPA/week (kcal/week)	2286 (±1922)	2112 (±1954)	2026 (±1614)	1	2048 (±1724)	2327 (±2112)		2357 (±1762)		0.448		2128 (±1886)		2263 (±1971)	2058 (±1665)	1

Continuous variables, X ± SD; categorical variables, %. BMI, Body mass index; CHO, carbohydrates; EBM, Estimated basal metabolism; PA, EEPA Energy expenditure in physical activity per week; SD, standard deviation; TCV, total caloric value; VFC visceral fat category; WC, waist circumference. Inactive, 0 times of physical activity performance per week. Active, one or more times of physical activity performance per week. *p*-values were obtained from one-way ANOVA for continuous variables, and Fisher exact test for categorical variables. *p*-values were adjusted by Bonferroni method (3 tests). Significance level *p* ≤ 0.05.

**Table 3 nutrients-12-01846-t003:** Associations between appetite level and waist circumference (cm), by *CLOCK* rs3749474 genotype.

Genotype		Model 1				Model 2				Model 3			
	*n*	β	(95 % CI)	*p*	*p**	β	(95 % CI)	*p*	*p**	β	(95 % CI)	*p*	*p**
C/C	203	3.1	0.3–5.8	0.087	0.001	3.0	0.1–5.8	0.120	<0.001	2.8	0.0–5.6	0.150	<0.001
C/T	185	7.1	4.3–9.8	<0.001		7.2	4.6–10.2	<0.001		7.3	4.6–10.1	<0.001	
T/T	49	15.7	7.9–23.5	<0.001		14.0	6.0–22.0	0.003		14.1	6.3–22.0	0.002	

The dependent variable was waist circumference (cm). Additive model. *p* values adjusted by model. *p*,* interaction. *p* and *p** values were adjusted by Bonferroni correction (3 test).

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
