# Peer review of "Polymorphic Appetite Effects on Waist Circumference Depend on rs3749474 CLOCK Gene Variant"

_nutrients, 2020, doi:10.3390/nu12061846_

Round 1

Reviewer 1 Report

Summary

The manuscript by Espinosa-Salinas et al. is rationalized, organized, and written reasonably well. The objective of the study is to investigate if genetic variation in the CLOCK gene modifies the effect of appetite on markers of metabolic diseases. The authors have a decent number of study subjects sufficient enough to investigate the hypothesis and followed appropriate methods. Their principal finding is that individuals with the minor allele of rs3749474 are at higher risk for an appetite-induced increase in waist circumference. This finding is clinically significant and contributes to our understanding of the genetic control of eating behavior and risk for metabolic diseases. Additional research in this area might help to advance the field of precision-nutrition.

Major Comments

  1. Statistically, there is a strong association between rs3749474 polymorphism and appetite-induced increase in waist circumference. Authors can demonstrate that association is biologically relevant by determining one or more markers related to appetite. These may include ghrelin, leptin, or other but appropriate hormones of their choice. It may not be feasible to determine these hormones in all of the study subjects considering their total number. They could randomly select some in each category of the genotype with sample size good enough to detect the differences. Adding this data renders the publication much more valuable.
  2. In many places, authors talk about chronobiology and CLOCK. It is true that the CLOCK gene regulates several aspects of chronobiology and that circadian rhythms regulate appetite. However, none of the variables in the study are analyzed temporally. It would be better not to emphasize too much on chronobiology.
  3. Appetite and waist circumference are the two major data variables involved in drawing conclusions from this manuscript. A more rigorous approach in determining these variables would be better. The authors do identify in their weaknesses that the visual analog scales are subjective. Other aspects can confound appetite quantification. Time since last meal, time of the day when appetite was quantified. Audiovisual cues/environment when appetite was measured. Who (a trained person or the subjects themselves) and where (at home or at a clinical center) was the appetite recording done? It should be mentioned whether the authors controlled such factors. If not, are the statistical models adjusted for these factors? Similarly, waist circumference – time since the last meal could distend the abdomen. Would abdominal distension affect waist circumference?
  4. Provide a rationale for downsizing the appetite categories from five to three. If this was an attempt to explore whether the associations were significant, that should be described.
  5. It would be interesting to see if any haplotypes are associated with the same variables that were investigated.
  6. The journal seems to conform to CONSORT guidelines. I do not see a flowchart of study subject recruitment. The authors should follow all the CONSORT guidelines.

Minor comments

Line 24: Full form of VAS

Line 96: what does “usual appetite” mean?

Line 104: Should it be 1 day/week instead of 1 day?

Line 108: “subjects” not “volunteers.”

Lines 103/104 and 108/109: Are these two different classifications or the same? If different, why?

Line 124: Were samples that failed in genotype calling redone?

Table 1:

1) Estimated Basal Metabolism (Kcal). Check units

2) Are simple sugars included in carbohydrates? Why separate them?

3) Be consistent with the terms “physical activity” and “exercise.” I prefer the first one.

Footnote:

  • The total number of participants was 442. How could the variables range from 437 to 487?
  • In the sentence “with age, sex, and genotyping data for at least some of the three polymorphisms.”, “some” should be a number.

General:

  • In the introduction, the authors could provide more details about the polymorphisms. A) What are the allele and genotype frequencies in the general population? B) Were any of these polymorphisms functional (affect CLOCK mRNA)?
  • Scientific information is clearly conveyed in the manuscript. But, a language editor can ease out the flow and iron out some minor issues. Or perhaps authors can themselves go through it one or two more times and correct it.

Author Response

We would like to thank you for your helpful comments with regard to submission of our paper entitled "Different appetite effects on waist circumference depend on rs3749474 CLOCK gene variant".

  1. Statistically, there is a strong association between rs3749474 polymorphism and appetite-induced increase in waist circumference. Authors can demonstrate that association is biologically relevant by determining one or more markers related to appetite. These may include ghrelin, leptin, or other but appropriate hormones of their choice. It may not be feasible to determine these hormones in all of the study subjects considering their total number. They could randomly select some in each category of the genotype with sample size good enough to detect the differences. Adding this data renders the publication much more valuable.

Thank you for your comments. Indeed, to examine relationships between the appetite questionnaires and biological markers such as ghrelin and leptin is a valuable consideration. Unfortunately, we have not blood samples for this purpose, which we will mention as a limitation in the discussion section.   

Line 238-341: Complement the current analyses by including appetite-related hormones such as ghrelin or leptin measurements would render the research much valuable, however, this could not be carried out due to insufficient blood sample.  

  1. In many places, authors talk about chronobiology and CLOCK. It is true that the CLOCK gene regulates several aspects of chronobiology and that circadian rhythms regulate appetite. However, none of the variables in the study are analyzed temporally. It would be better not to emphasize too much on chronobiology.

Thank you very much for your comment. Actually, although the CLOCK gene regulates aspects of chronobiology, in this study no temporality has been analyzed. We have proceeded to remove or tone down this point across the revised manuscript.

  1. Appetite and waist circumference are the two major data variables involved in drawing conclusions from this manuscript. A more rigorous approach in determining these variables would be better. The authors do identify in their weaknesses that the visual analog scales are subjective. Other aspects can confound appetite quantification. Time since last meal, time of the day when appetite was quantified. Audiovisual cues/environment when appetite was measured. Who (a trained person or the subjects themselves) and where (at home or at a clinical center) was the appetite recording done? It should be mentioned whether the authors controlled such factors. If not, are the statistical models adjusted for these factors? Similarly, waist circumference – time since the last meal could distend the abdomen. Would abdominal distension affect waist circumference?

To clarify this aspect, a more rigorous description in determining these two variables (waist circumference and appetite) has been added. With regard to waist circumference (and all anthropometric parameters), it was measured by trained nutritionists on an empty bladder with a minimum of two hours fasting. We have added to manuscript the following text:

Line 102-104: Waist and hip circumferences were determined using a Seca 201 non-elastic tape (Quirumed, Valencia, Spain). All anthropometric parameters were measured by trained nutritionists on an empty bladder with a minimum of two hours fasting, as described elsewhere (Bibilonic et al., 2019).

Furthermore, concerning appetite quantification, a Likert scale with 5 points has been used. This question was asked by trained nutritionist always at the research center and when the subjects had been fasting for at least two hours following controlled guidelines as previously applied (Harris et al., 2008; Antonini et al., 2014).  We have added to manuscript the following text:

Line 105-108: To measure appetite degree, a Likert scale with 5 points has been used (where 1-2 means a low rate of appetite, 3 an intermediate level and 4-5 a lot of appetite). This question was asked by trained nutritionist always at the research center and when the subjects had been fasting for at least two hours following published protocol (Harris et al., 2008; Antonini et al., 2014).

  1. Provide a rationale for downsizing the appetite categories from five to three. If this was an attempt to explore whether the associations were significant, that should be described.

Thank you very much four your comment. Since it is a semi-quantitative method, we decided to group appetite in 3 categories: low, medium and high, in order to optimize the statistical analyses. The points 1 and 2 on the scale were considered low appetite, 3 point intermediate appetite and the points 4 and 5 were considered high appetite. We have added to manuscript the following text:

Line 111-112: Then, since it is a semi-quantitative method the data were merged in 3 categories in order to optimize the statistical analysis. This scale was categorized in low appetite, medium appetite and high appetite with 1-2, 3, and 4-5 points in the scale, respectively.

  1. It would be interesting to see if any haplotypes are associated with the same variables that were investigated.

The analysis of the CLOCK gene and the different SNPs was performed separately. The idea of haplotypes turns out to be very interesting but since the n is relatively small, we have not done it as it would limit the statistical performance.  However, there is previous evidence about the haplotype, rs3749474, rs45807041, and rs1801260 (3111TC) which showed that carriers of the haplotype CGA had a lower BMI, weight, waist circumference, adiponectin concentration, and blood pressure than did noncarriers (Garaulet et al., 2009). Taking into account these observations, we have added in the discussion section the following test:

Line 278-282: In turn, scientific literature has shown that in the analysis by haplotypes rs3749474, rs45807041 and rs1801260, carriers of the haplotype CGA had a lower BMI, weight, waist circumference than did noncarriers [11]. This evidence corroborates our results and in turn indicates the possibility that the other polymorphisms (rs45807041 and rs1801260) affect body weight and waist diameter together.

  1. The journal seems to conform to CONSORT guidelines. I do not see a flowchart of study subject recruitment. The authors should follow all the CONSORT guidelines.

Thank you very much four your comment. According the CONSORT guidelines, we attached the flowchart of study as supplementary material (For the reviewer: Attached to the end of the manuscript. Line 488).

Furthermore, we described it in the materials and methods section as follows:

Line 87-92: A total of 1249 subjects were contacted, where 954 were recruited. Of these, a total of 557 participants were finally selected. Due to uncompleted questionnaires, 487 samples were left for analysis. The sample size of the variables ranges from 437 to 487 due to some missing data. No imputation was performed. The sample size was shown considering all subjects with age, sex and genotyping data for at least some one of the three polymorphisms (n = 442). CONSORT flow diagram is shown as supplementary material.

Minor comments

  • Line 24: Full form of VAS.

We have specified the question

Line 24: Participants were classified according to their appetite feelings with a LIKERT scale.

  • Line 96: what does “usual appetite” mean?

The term "habitual appetite" is colloquially used in Spanish language to refer how do you feel about your eating and feeding status. Therefore, this term was used in the survey of the subjects. In order to clarify this issue, we have added the following test:

Line 110-111: The participants responded according to their ”habitual appetite status”, with a commonly used question by Spanish population to refer to how you feel about your eating and feeding.

  • Line 104: Should it be 1 day/week instead of 1 day?

Indeed, it is 1 day/week and it has been changed accordingly in the manuscript.

Line 125: Those subjects that reported at least 1 day/week of physical activity were considered active.

  • Line 108: “subjects” not “volunteers.”

The word "volunteers" has been replaced by "subjects".

Line 128: Moreover, the Minnesota Leisure Time Physical Activity Questionnaire (MLTPAQ) was administered to quantitatively measure the average physical activity practice (kcal/day) by the subjects according to published criteria [23,24]

  • Lines 103/104 and 108/109: Are these two different classifications or the same? If different, why?

These classifications are different. Exercise levels have been applied to assess the frequency of exercise, considering it as an increase in body mobility being a planned and repetitive activity. Minnesota Leisure Time Physical Activity Questionnaire has been applied to assess the amount of physical activity in (kcal/day), considering physical activity as any voluntary movement produced by the muscles and resulting in the expenditure of energy. We have added this information to the manuscript:

Line 114-118: In order to evaluate exercise levels, it was assessed the frequency of exercise, considering it as an increase in body mobility being a planned and repetitive activity (Espinosa-Salinas et al., 2019). The Minnesota Leisure Time Physical Activity Questionnaire has been applied to assess the amount of physical activity in (kcal/day), considering physical activity as any voluntary movement produced by the muscles and resulting in the expenditure of energy (Espinosa-Salinas et al., 2019).

  • Line 124: Were samples that failed in genotype calling redone?

The samples were made in duplicate, resulting than more 99% of the genotyping results were valid. We added this information in the manuscript.

Line 145-146: The samples were made in duplicate, resulting more 99% of the genotyping results were technically valid.

Table 1:

  • Estimated Basal Metabolism (Kcal). Check units.

Thank you very much for your comment. Estimated Basal Metabolism units have been checked and changed to Kcal/day.

2) Are simple sugars included in carbohydrates? Why separate them?

Indeed, simple sugars were included in the total carbohydrates. However, we considered separating out simple sugars since they may show a different outcome than overall carbohydrates given their role on sweetness and satisfaction and eventually on appetite (Ferreti et al., 2017).

Line 243-245: Simple sugars were included in the total carbohydrates. However, we considered separating out simple sugars since they may show a different outcome than overall carbohydrates given their role on sweetness and satisfaction as well as potentially on appetite (Ferreti et al., 2017).

3) Be consistent with the terms “physical activity” and “exercise.” I prefer the first one.

In agreement to the reviewer's comment, it has been considered physical activity and exercise as two different terms. Exercise levels have been applied to assess the frequency of exercise, considering it as an increase in body mobility being a planned and repetitive activity (Espinosa-Salinas et al., 2019). Minnesota Leisure Time Physical Activity Questionnaire has been applied to assess the amount of physical activity in (kcal/day), considering physical activity as any voluntary movement produced by the muscles and resulting in the expenditure of energy (Espinosa-Salinas et al., 2019). We have added this information to the manuscript:

Line 114-118:  In order to evaluate exercise levels, it was assessed the frequency of exercise considering it as an increase in body mobility being a planned and repetitive activity (Espinosa-Salinas et al., 2019). The Minnesota Leisure Time Physical Activity Questionnaire has been applied to assess the amount of physical activity in (kcal/day), considering physical activity as any voluntary movement produced by the muscles and resulting in the expenditure of energy (Espinosa-Salinas et al., 2019). 

Footnote:

  • The total number of participants was 442. How could the variables range from 437 to 487?

Thank you very much four your comment. Of the total number of subjects analysed (n = 487), finally the sample size of the variables ranges from 437 to 487 due to some missing data. For this reason, sample size was set up considering all subjects with age, sex and genotyping data for at least some one of the three polymorphisms (n = 442). We have added accurate information in the results section as well as in the new flow chart:

Line 161-163: A total of 442 (329 women and 113 men, with mean and ± SD age of 37.63 ±12.38 years) subjects were analyzed, considering the sample size, which included all subjects with age, sex and genotyping data for at least one of the three polymorphisms.  

  • In the sentence “with age, sex, and genotyping data for at least some of the three polymorphisms.”, “some” should be a number.

We have changed the text according your comment:

Table 1: The sample size was defined considering all subjects with age, sex and genotyping data for at least one of the three polymorphisms.

General:

  • In the introduction, the authors could provide more details about the polymorphisms. A) What are the allele and genotype frequencies in the general population? B) Were any of these polymorphisms functional (affect CLOCK mRNA)?

In agreement to the reviewer's comment, we have added the following in the introduction section:

Line 63-65: The allele and genotype frequencies of rs3749474 are C: 66%, T: 34% and C/C: 43% C/T: 45% T/T: 12% in the European population (Ensembl GRCh38).

Line 65-66: In turn, polymorphisms located within the 3′-UTR of the mRNA, can affect the functionality of the mRNA, as seen in the case of rs3749474 and rs1801260 (Garaulet et al., 2010; Xu et al., 2010 ; Garaulet et al., 2010)

Scientific information is clearly conveyed in the manuscript. But, a language editor can ease out the flow and iron out some minor issues. Or perhaps authors can themselves go through it one or two more times and correct it.

An effort has been made to improve English language, which have checked by English native.

Reviewer 2 Report

The authors present a cross-sectional analysis on the effects of SNPs in the CLOCK gene on anthropometric measures with and without interaction with dietary parameters.

The introduction is sufficient.

The methods need clarification which anthropometric measures have been tested for an interaction effective with appetite. The authors present data on waist circumference, the only parameter with significant results, but state that others were tested, too. Bonferroni correction needs to be applied to number of SNPs, number of interaction tests with dietary factors and number of tested anthropometric parameters in all tables on results.

This critique also applies to the confirmation analysis, which was done in another data set and for which more details on cohort size and cohort structure should be given.

Furthermore, Table 3 and Fig. 1 indicate, that heterozygous CT subjects, too, show a significant interaction with appetite. Therefore, rather than a dominant model of the T allele, it appears, that an additive model should be applied.

In general, please use a consistent amount of decimals for all variables, always corresponding to the precision of the respective instruments (e.g. waist circumference: 1 decimal, only).

Discussion may need revision, if the new Bonferroni correction leads to elimination of significance.

Author Response

We would like to thank you for your helpful comments with regard to submission of our paper entitled "Different appetite effects on waist circumference depend on rs3749474 CLOCK gene variant".

  • The methods need clarification which anthropometric measures have been tested for an interaction effective with appetite. The authors present data on waist circumference, the only parameter with significant results, but state that others were tested, too. Bonferroni correction needs to be applied to number of SNPs, number of interaction tests with dietary factors and number of tested anthropometric parameters in all tables on results.

Thank you for your relevant comment concerning statistical corrections for multiple comparisons. Indeed, we only analysed body mass index, fat mass, visceral fat and waist circumference as a priori driven-hypothesis anthropometric measurements related with appetite for interactions.  Then, we applied the Bonferroni correction procedure, when we achieved the statistical results as reported in the initial manuscript, including the confirmatory sub-study. Therefore, it was not necessary to perform further correction analyses involving multiple comparisons. The unadjusted p values were: p = 0.00036 (Model 1), p = 0.00019 (Model 2) and p = 0.000095 (Model 3). In case of making the correction taking into account the rest of the dependent variables, p adjusted interaction values continued to be equally significant (Model 1: p = 0.004; Model 2: p = 0.002 Model 3: p = 0.001).  We have commented this analysis in the discussion section.

Line 283-287: The adjustment for multiple comparisons has taken into account 3 SNPs * 1 Appetite interaction test.  The unadjusted p values were: p = 0.00036 (Model 1), p = 0.00019 (Model 2) and p = 0.000095 (Model 3). In case of making the correction taking into account the rest of the dependent variables, p adjusted interaction values continued to be equally significant (Model 1: p = 0.004; Model 2: p = 0.002 Model 3: p = 0.001).

This critique also applies to the confirmation analysis, which was done in another data set and for which more details on cohort size and cohort structure should be given.

About the confirmatory analysis, Bonferroni correction was applied with 81 tests since 9 SNPs and 9 interaction variables were taken into account in the study. In this analysis, the dependent variables were not considered for the correction, since the SNPs tested and the number of interactions, were sufficient. Failure to address multiple comparisons appropriately can introduce excess false positive results and make subsequent studies following up those results inefficient (Gao et al., 2011). This situation can be the case for GWAS analysis, where the correction applies only to the number of SNPs (Donaldson et al., 2015. pp. 148-149). Furthermore, we only analysed other dependent variables as a priori driven-hypothesis anthropometric measurements related with appetite for interactions. However, we have commented on this in the discussion.

Line 287-291: Although we had a greater number of adjustments in the confirmatory analysis (81 tests), in this case the dependent variables were not taken into account since we have considered not necessary to perform further correction analyses involving multiple comparisons. Failure to address multiple comparisons appropriately can introduce excess false positive results and make subsequent studies following up those results inefficient (Gao et al., 2011).

  • Discussion may need revision, if the new Bonferroni correction leads to elimination of significance.

We have added the comments in the above section in the discussion, although the new statistical consideration did not change the previously reported outcomes.

  • Furthermore, Table 3 and Fig. 1 indicate, that heterozygous CT subjects, too, show a significant interaction with appetite. Therefore, rather than a dominant model of the T allele, it appears, that an additive model should be applied.

Thank you very much four your comment. Indeed, it is an additive model as shown in Table 3 and Fig. 1. We have added several comments concerning the model throughout the text to further clarify this point:

Line178-179: Among the three genetic models evaluated (additive, dominant and codominant), the best fit was obtained with the additive model (in which each copy of T modifies the risk by an additive amount and therefore T/T homozygotes have twice the risk of C/T heterozygotes).

Line 212: With regards to anthropometric measurements, a relevant finding that emerged from this study is a directly proportional association between the minor T allele of the CLOCK rs3749474 variant and the waist circumference, for each increase in the grade of appetite (additive model).

Line 270-272: Interestingly, our results have also found an interaction concerning to waist circumference: carriers of the T/T genotype increased by about 14 cm for each increase in the appetite rate (C/T: 7cm and C/C: 3 cm).

Line 276: This outcome explains that under an additive model, subjects with T/T genotype may be at increased risk for the etiopathogenesis of diseases related to metabolic syndrome and abdominal obesity (and this risk seems to grow with each increase in the level of appetite according to the results found).

  • In general, please use a consistent amount of decimals for all variables, always corresponding to the precision of the respective instruments (e.g. waist circumference: 1 decimal, only).

Thank you very much four your comments. We have improved the accuracy of the variables by adding decimals corresponding to the precision of the respective instruments such as fat and lean mass.

Round 2

Reviewer 1 Report

Authors addressed my comments sufficiently.

Author Response

We greatly appreciate the reviewers for their complimentary comments and suggestions. We have reviewed the English language again to check for spelling. We enclose the corrected version.

Reviewer 2 Report

Thanks for this thorough and detailed revision and reply, which answered and addressed all points of criticism sufficiently.

As a resulting and thus remaining minor point, the abstract should now be revised in order to not only mention the additive model, but also to report the respective effect as an additive effect (not: "minor genotype x increment of degree of appetite"; but: "risk allele x increment of degree of appetite").

Author Response

We greatly appreciate the reviewers for their complimentary comments and suggestions. Certainly, the change indicated is much more appropriate. We have added "risk allele" in the abstract.

Line 27-30: Additive model analysis (adjusted by age, gender, exercise and energy intake) showed that risk allele carriers, increased the waist circumference around 14 cm (β = 14.1, CI = 6.3 – 22.0) by each increment in the level of appetite.

We have reviewed the English language again to check for spelling. We enclose the corrected version. 
